# Physical Activity and Sedentary Time in Pregnancy: An Exploratory Study on Oxidative Stress Markers in the Placenta of Women with Obesity

**DOI:** 10.3390/biomedicines10051069

**Published:** 2022-05-05

**Authors:** Saghi Zafaranieh, Anna M. Dieberger, Barbara Leopold-Posch, Berthold Huppertz, Sebastian Granitzer, Markus Hengstschläger, Claudia Gundacker, Gernot Desoye, Mireille N. M. van Poppel

**Affiliations:** 1Institute of Human Movement Science, Sport and Health, University of Graz, 8010 Graz, Austria; saghi.zafaranieh@uni-graz.at; 2Department of Obstetrics and Gynecology, Medical University of Graz, 8036 Graz, Austria; anna.dieberger@medunigraz.at (A.M.D.); barbara.leopold-posch@uniklinikum.kages.at (B.L.-P.); gernot.desoye@medunigraz.at (G.D.); 3Division of Cell Biology, Histology and Embryology, Gottfried Schatz Research Center, Medical University of Graz, 8010 Graz, Austria; berthold.huppertz@medunigraz.at; 4Institute of Medical Genetics, Center for Pathobiochemistry and Genetics, Medical University of Vienna, 1090 Vienna, Austria; sebastian.granitzer@meduniwien.ac.at (S.G.); markus.hengstschlaeger@meduniwien.ac.at (M.H.); claudia.gundacker@meduniwien.ac.at (C.G.); 5Karl-Landsteiner Private University for Health Sciences, 3500 Krems, Austria

**Keywords:** obesity, placenta, oxidative stress, physical activity, sedentary behavior

## Abstract

Regular moderate-to-vigorous physical activity (MVPA) and reduced sedentary time (ST) improve maternal glucose metabolism in pregnancy. More MVPA and less ST outside pregnancy increase antioxidant capacity, hence, are beneficial in preventing oxidative stress. The placenta is the first line of defense for the fetus from an adverse maternal environment, including oxidative stress. However, effects of MVPA and ST on oxidative stress markers in the placenta are unknown. The purpose of this study was to assess the association of MVPA and ST in pregnancy with oxidative stress markers in placentas of overweight/obese women (BMI ≥ 29 kg/m^2^). MVPA and ST were objectively measured with accelerometers at <20 weeks, 24–27 and 35–37 weeks of gestation. Using linear Bayesian multilevel models, the associations of MVPA and ST (mean and changes) with mRNA expression of a panel of 11 oxidative stress related markers were assessed in 96 women. MVPA was negatively correlated with HSP70 mRNA expression in a sex-independent manner and with GCLM expression only in placentas of female fetuses. ST was positively associated with HO-1 mRNA expression in placentas of male neonates. None of the other markers were associated with MVPA or ST. We speculate that increasing MVPA and reducing ST attenuates the oxidative stress state in placentas of obese pregnant women.

## 1. Introduction

Obesity is increasingly prevalent in women of reproductive age and has been recognized as a risk factor for a number of maternal and fetal complications including preterm birth, hypertension, gestational diabetes mellitus (GDM), and preeclampsia [1,2,3]. Adipose tissue is metabolically highly active, acting as an endocrine organ [4]. In pregnancy, obesity promotes a maternal environment with increased pro- and anti-inflammatory markers [5]. The placenta is exposed to both maternal and fetal blood circulation and, hence, can be a mediator between the maternal environment and fetal outcome. It is exposed to the maternal environment and may protect the fetus from adverse effects of maternal obesity, among other pregnancy conditions.

In addition to its inflammatory condition, maternal obesity is associated with oxidative and nitrative stress [6,7]. Oxidative stress signifies an augmented reactive oxygen species (ROS) state not being scavenged by the antioxidative capacity of tissues, i.e., exceeding antioxidative capacity of the tissue [8]. ROS are hyperreactive molecules resulting from the reduction of molecular oxygen and are primarily formed by mitochondrial oxidative phosphorylation. Some of the most commonly known ROS are superoxide (O_2_•−), hydroxide (OH−•), and hydrogen peroxide (H_2_O_2_). Excess ROS and reactive nitrogen species can cause cellular damage and tissue malfunction and, thus, harm placental development [9]. However, at physiological levels, ROS regulate gene transcription and downstream processes such as trophoblast proliferation, inflammation, vasodilation, and angiogenesis [10].

Over the years, evidence on placental and fetal changes associated with maternal obesity and diabetes has accumulated [6,11]. Elevated levels of maternal circulating fatty acids and lipids in the placenta create a lipotoxic environment which affects placental inflammation, oxidative stress, and function [12]. As maternal adiposity and body mass index (BMI) increase, there is excessive ROS production in the placenta [13], and a rise in risk of associated fetal and long term maternal disorders such as GDM [14] and cardiovascular disease [15], indicating the importance of weight and obesity management before and during pregnancy. While weight loss including medications to lose weight are not recommended during pregnancy, lifestyle changes focusing on healthy diet, regular physical activity (PA), and decreasing sedentary time (ST) are key for a healthy pregnancy.

PA and ST are distinct behaviors with independent effects on health outcomes, such as insulin sensitivity and fetal adiposity [16]. While PA can reduce potential complications for mother and fetus [17], less is known about the effects of ST. The American College of Obstetrics and Gynecologists (ACOG) and the American College of Sports Medicine (ACSM) recommends all women without contraindications to be physically active throughout their pregnancy and advise to participate in at least 150 min of moderate intensity PA per week [18,19], but include no recommendations regarding ST.

Outside of pregnancy, regular PA increases the antioxidant capacity [20]. In pregnancy, PA and exercise training can improve placental development and reduce pro-inflammatory markers in maternal blood [21,22,23,24]. Moreover, combined aerobic and resistance training during the second half of pregnancy decreased components of the oxidative stress system in the placenta. Mitochondrial superoxide and hydrogen peroxide levels were lowered by 8% and 37%, respectively [25]. Sedentary behavior is associated with neonatal adiposity [16]. However, little is known about the effect of PA or ST on the placenta, except for moderate-to-vigorous PA (MVPA) induced changes in placental transcriptome [26] and an ST induced lower expression of placental genes linked to lipid transport [27].

The purpose of this study was to assess the association of PA as well as ST in pregnancy with oxidative stress markers in placentas of obese women. PA and ST are often semi-quantified by self-administered questionnaires, which are subject to bias [28]. To overcome this problem, we employed objective measurements of PA and ST. The placenta’s capacity to mount adaptive responses to continuous exposure of adverse stressors requires multiple measurements of PA and ST throughout pregnancy. Therefore, we measured PA and ST at three time points in pregnancy. This also enabled us to investigate the association of changes in PA and ST during pregnancy on the outcomes. The placental targets were chosen, because of their established role in regulating stress response or as marker of oxidative stress.

## 2. Materials and Methods

### 2.1. DALI Study

This is a secondary analysis of the vitamin D and lifestyle intervention for GDM prevention (DALI) multicenter randomized controlled trial study conducted between 2012 and 2015 in 11 study centers in nine European countries (Austria, Belgium, Denmark (Odense, Copenhagen), Ireland, Italy (Pisa, Padua), The Netherlands, Poland, Spain, and United Kingdom). The study was registered under trial registration number ISRCTN70595832 and was approved by all local ethics committees.

Pregnant women <20 weeks of gestation with a singleton pregnancy, aged ≥18 years with a pre-pregnancy BMI of ≥29 kg/m^2^ were invited to participate. Women with GDM at baseline based on International Association of Diabetes and Pregnancy Study Group (IADPSG) criteria (fasting venous plasma glucose ≥5.1 mmol/L and/or 1-h glucose of ≥10 mmol/L and/or 2-h glucose of ≥8.5 mmol/L) [29] or who were pre-diagnosed with diabetes, chronic medical conditions, or psychiatric disorders were excluded. Other exclusion criteria were an inability to walk 100 m safely, requirement for a complex diet and not being able to communicate with the lifestyle coach due to a lack of language proficiency. To allow for longitudinal analyses, only those with at least two out of three physical activity measurements were included in this study.

After written informed consent, women were randomized into four groups receiving counselling for Healthy Eating (HE), Physical Activity (PA), Healthy Eating + Physical Activity (HE + PA), and a control group receiving usual care (UC) in the lifestyle trial. For this analysis, all randomized participants were combined into one cohort and analyzed observationally.

### 2.2. Data Collection

Data were collected from participants at four time points: baseline (<20 weeks), at 24–28 weeks, at 35–37 weeks and after delivery. Maternal data collection included anthropometrics (weight, height), blood samples and questionnaire data (including age, ethnicity, parity, and smoking status). Additional information on pre-pregnancy weight, maternal/paternal smoking, alcohol consumption, and medical history was collected by questionnaire. Data on birth outcomes were collected from medical files.

### 2.3. Physical Activity and Sedentary Time

Physical activity was measured objectively by accelerometers (ActiGraph GT1M, GT3X+ or Actitrainer; Pensacola, FL, USA) at <20 weeks, 24–28 weeks, and 35–37 weeks. Women were asked to wear an accelerometer positioned over the right hip for at least 3 days during waking hours and remove the device only when swimming or showering and indicate the reason and duration of the removal in their activity records. If the zero-count period exceeded 90 min, it was considered non-wear time. For being included in the analysis, at least 3 valid full day measurements per time period with a daily wear time of >480 min was required. Freedson cut-off points were applied to define the average time spent sedentary (<100 counts/min), in light (100–1951 counts/min), and moderate-to-vigorous physical activity (MVPA) (>1951 counts/min) [30]. Swimming time was added to minutes spent in MVPA [31]. For analyses, sedentary time was calculated as percentage of total daily accelerometer wear time (% ST).

### 2.4. Placental Tissue Collection and RNA Isolation

Within 30 min of delivery, placental tissue was dissected from the maternal and fetal side of four quadrants and stored in cryotubes filled with RNA-later at −20 °C. Upon arrival of the samples in the central lab of Graz, RNA-later was removed and two pieces of each approx. 20 mg were pooled from both sides. Tissue samples were homogenized using the MagNA Lyser Instrument (Roche Diagnostics, Vienna, Austria: 2–3 runs, 6500 rpm, 20 s). miRNeasy Mini Kit (Qiagen, Hilden, Germany, #217004) was used according to the manufacturer’s instruction for RNA isolation. QIAxpert (Qiagen) and Agilent 2100 Bioanalyser systems (Agilent, Santa Clara, CA, USA) were utilized for determination of RNA concentration and quality.

The expression of specified mRNAs (heme oxygenase 1 (HO-1), growth arrest and DNA damage inducible alpha (GADD45a) and beta (GADD45b), nuclear factor erythroid 2-related factor 2 (Nrf2), lipoprotein-associated phospholipase A₂ (Lp-PLA2), metallothionein 2A (MT2A), and heat shock protein 70 (HSP70)) were quantified by molecular counting using NanoString nCounter Analysis Technology (Nanostring Technologies, Seattle, WA, USA). Nrf2 is a transcription factor acting as master regulator of its downstream effectors HSP, HO, GADD45, and Lp-PLA2 [32]. Components of the glutathione system are ubiquitous key actors in the intracellular oxidative stress systems. The readouts of signal intensities are given in arbitrary units (AU). The probes for the investigated genes were part of a customized CodeSet (nCounterTM PlexSetTM) with a total of 24 probes, including probes for three validated housekeeping genes (ornithine decarboxylase antizyme 1 (OAZ1), WD repeat-containing protein 45-like (WDR45L), and tata-box-binding protein (TBP)), which were used for hybridization of a total of 490 ng RNA per sample according to the manufacturer’s protocol. Quality control and normalization was done utilizing the NanoString nSolver Analysis Software v4.0 (NanoString Technologies, Seattle, WA, USA).

Glutamate-cysteine ligase modifier subunit (GCLM), glutamate-cysteine ligase catalytic subunit (GCLC), glutathione peroxidase 1 (GPX1), and glutathione reductase (GR) were quantified by RT-qPCR. The cDNA was synthesized using the reverse transcriptase enzyme method. Total RNA (1µg per sample) was transcribed using the qScript cDNA Synthesis Kit (Quantabio, Beverly, MA, USA, #95047) following the manufacturer’s instructions. Two µL cDNA solution was used as template in a 15 µL gene expression assay reaction, following Applied Biosystems StepOnePlus Real-Time PCR System protocol. The employed primers were Hs00824723_m1 (UBC = Ubiquitin C), Hs00427620_m1 (TBP), Hs00892604_m1 (GCLC), Hs00978072_m1 (GCLM), Hs00167317_m1 (GR), and Hs00829989_gh (GPX1). UBC and TBP were used as reference genes, and non-amplification control as negative control.

### 2.5. Statistical Methods

Participant characteristics are presented by mean and standard deviation (SD), median and interquartile range (IQR) or count and proportion. The characteristics of included and excluded participants were compared using unpaired *t*-test and chi-square test. Accelerometer data (%ST, MVPA) are presented separately for each time point. Differences between time points are tested by paired sample *t*-tests.

To analyze the relationship between the repeatedly measured MVPA and ST and placental markers, a two-step analysis was applied [33]. First, a multilevel linear model was performed with the gestational age at the accelerometer measurements as independent variable and the repeated MVPA or ST measurements as dependent variable. A two-level model including a random intercept and random slope was modelled with the repeated measurements on level one, nested within each individual on level two. Thereby, individual slopes representing changes in MVPA and ST per gestational week could be estimated for each participant. These individual slopes were then exported as new variables, indicating estimated individual changes in MVPA and ST in minutes or % accelerometer wear time per gestational week. Additionally, individual mean values of MVPA and ST were calculated for each participant, representing the average time spent sedentary or in MVPA during pregnancy. To make estimates easier to interpret, mean MVPA and ST were transformed into units of 10 min and 10% wear time, respectively, and changes in 10 min per 4 weeks or 10% per 4 weeks, respectively, for the slope variables.

Secondly, an additional multilevel linear analysis was performed, using the newly created mean and slope variables of MVPA and ST as independent variables and the different placental markers as dependent variables. The MVPA and ST variables were added in the same model, to be able to estimate associations independent of each other. Again, a two-level structure allowing for a random intercept was applied, with individuals on level one nested within the different DALI study centers (n = 8; at three centers (Odense, Pisa, UK) no placenta tissue was sampled) on level two. The models were additionally adjusted for the covariates maternal smoking (yes vs. no), educational level (high vs. low), ethnicity (European vs. non-European descent), pre-pregnancy BMI (continuous, log-transformed), gestational age at birth (weeks, continuous), parity (nulliparity vs. multiparity), birth mode (caesarean section vs. spontaneous delivery), offspring sex (male vs. female), and randomization for the HE intervention (HE vs. no HE).

Effect modification by offspring sex was considered by adding interaction terms of the MVPA and ST variables and sex to each model. If significant effect modification (based on 90% credible intervals) was found, models were subsequently stratified by sex.

As sensitivity analyses, GDM diagnosis and prostaglandin use [27] during delivery were separately added to the models as potential confounders to inspect whether these variables influence the relationship between maternal MVPA and ST and the placental markers.

Due to the relatively small number of participants, instead of a maximum likelihood procedure, linear Bayesian multilevel models were used for the two-step approach. A non-informative flat prior was used. Diagnostics for model fit and convergence included effective sample size, autocorrelation, R-hat and comparisons of the observed and predicted posterior distribution and were inspected individually for each model. The distribution of the response variable per default is a Gaussian distribution. Alternatively, if model fit was not satisfactory, a skewed normal distribution was chosen. All model estimates are presented with a 95% credible interval.

Descriptive analysis and comparison of characteristics were analyzed with IBM SPSS Statistics (version 27.0) [34]. The main analysis of the study was performed in R; a language and environment for statistical computing (version 4.0.4) [35]. Bayesian multilevel analyses were performed using the brms package (version 2.16.1) [36], which is a front end for Stan (version 2.21.2) [37]. Plots were produced with ggplot2 (version 3.3.5) [38] and GraphPad Prism software (version 9.2.0).

## 3. Results

The flow chart of participants and reasons of exclusion are shown in Figure 1. Of 192 placental tissues dissected from randomized participants, RNA expressions were successfully quantified from 184 participants. Among these participants, 96 women had accelerometer measurements from at least two time points and were included in the longitudinal study. The characteristics of the study sample at baseline are summarized in Table 1. Most participants were Caucasian, non-smoking women with high education, on average 33.3 ± 5.3 years old and with a median pre-pregnancy BMI of 32.9 (IQR 28.7–45.0) kg/m^2^. Most women (68.8%) delivered spontaneously. Women included in the analyses were significantly older and more often had a spontaneous delivery compared to those excluded.

### 3.1. Physical Activity and Sedentary Time throughout Pregnancy

Average MVPA and % ST at three time points in pregnancy are summarized in Table 2. MVPA decreased significantly in this sample of pregnant women from <20 weeks to 35–37 weeks (*p* = 0.003). In contrast, % ST increased significantly from <20 weeks towards 24–28 weeks (*p* = 0.026) (Table 2). The estimated individual slopes of MPVA (Figure 2a) and % ST (Figure 2b) throughout pregnancy also indicate that most women had a decrease in MPVA and increase in %ST per gestational week.

### 3.2. Association of MVPA and ST with Oxidative Markers in the Placenta

The associations of mean MVPA, mean % ST, changes in MVPA and changes in % ST in pregnancy with the oxidative stress markers are presented in Table 3. Mean MVPA and ST are, respectively, reported in 10 min and 10% units. The changes (slopes) in MVPA and ST are shown in 10 min and 10% changes per four weeks.

A higher mean MVPA was associated with lower HSP70 and GCLM mRNA expression in placenta (−0.02 AU [95% CI −0.05, −0.002] and -0.04 AU [95% CI −0.08, −0.01], respectively) (Figure 3a). Changes in MVPA or % ST throughout pregnancy were not associated with any of the other placental outcomes.

Adding GDM or prostaglandin use (yes/no) to the models in sensitivity analyses did not change the results meaningfully.

### 3.3. Sex-Specific Associations of MVPA and ST with Oxidative Markers in the Placenta

To identify potential sex-specific responses, all associations were tested for interactions with neonatal sex. These analyses found an interaction of % ST with neonatal sex for HO-1, but not for other mRNA outcomes. When the association was assessed in both sexes separately, a significant positive association between HO-1 expression and mean % ST (0.21 AU [95% CI 0.03, 0.39]) (Figure 3b) and changes in % ST (5.84 AU [95% CI 1.85, 9.55]) was only seen in placentas of male neonates.

**Table 3 biomedicines-10-01069-t003:** The longitudinal association of mean moderate-to-vigorous physical activity (MVPA), changes in MVPA, mean sedentary time (ST) (10% wear time), and changes in ST (slope; 10% wear time/4 weeks) with mRNA expressions in the placenta of obese pregnant women.

	Mean MVPA (10 min) Estimate (95% CI)	Change (Slope) MVPA (10 min/4 Weeks) Estimate (95% CI)	Mean % ST (10% Wear Time) Estimate (95% CI)	Change (Slope) % ST (10% Wear Time/4 Weeks) Estimate (95% CI)
Nrf2, AU	−0.02 (−0.05, 0.01)	0.54 (−0.87, 1.98)	−0.002 (−0.07, 0.08)	0.97 (−0.63, 2.57)
MT2A, AU	−0.03 (−0.12, 0.06)	2.33 (−2.15, 6.55)	0.10 (−0.15, 0.35)	2.91 (−2.16, 8.00)
GADD45a, AU	0.03 (−0.03, 0.09)	−0.35 (−3.48, 2.76)	0.15 (−0.03, 0.34)	1.27 (−2.19, 4.83)
GADD45b, AU	−0.01 (−0.09, 0.06)	0.63 (−3.14, 4.48)	0.10 (−0.11, 0.32)	−0.67 (−4.74, 3.67)
HO-1, AU	0.01 (−0.04, 0.06)	0.78 (−1.73, 3.33)	0.02 (−0.13, 0.16)	2.37 (−0.53, 5.27)
Male (n = 50)	0.01 (−0.05, 0.07)	0.58 (−2.90, 3.90)	**0.21 (0.03, 0.39)**	5.84 (1.85, 9.55)
Female (n = 43)	0.034 (−0.05, 0.12)	1.48 (−2.00, 5.14)	−0.05 (−0.27, 0.17)	3.58 (−0.78, 7.88)
HSP70, AU	**−0.02 (−0.05, −0.002)**	−0.78 (−1.85, 0.30)	0.01 (−0.06, 0.07)	0.88 (−0.29, 2.06)
Lp-PLA2, AU	0.10 (−0.02, 0.23)	0.58 (−6.00, 7.04)	0.30 (−0.05, 0.66)	−2.46 (−9.91, 4.87)
GCLM, AU	**−0.04 (−0.08, −0.01)**	0.21 (−1.62, 1.84)	−0.04 (−0.15, 0.07)	0.64 (−1.15, 2.19)
GR, AU	−0.001 (−0.02, 0.01)	−0.35 (−0.93, 0.25)	−0.01 (−0.03, 0.03)	0.24 (−0.40, 0.85)
GPX1, AU	−0.14 (−0.74, 0.44)	−6.37 (−31.73, 21.13)	0.17 (−1.34, 1.78)	3.07 (−23.83, 30.50)
GCLC, AU	−0.002 (−0.02, 0.02)	0.60 (−0.26, 1.47)	−0.01 (−0.06, 0.05)	−0.56 (−1.49, 0.32)

AU = arbitrary unit, CI = credible interval, Nrf2 = Nuclear factor erythroid 2-related factor 2, MT2A = Metallothionein 2A, GADD45a = Growth Arrest and DNA Damage Inducible Alpha, GADD45b = Growth Arrest and DNA Damage Inducible Beta, HO-1 = Heme oxygenase 1, HSP70= Heat Shock Protein 70, LP_PLA2 = Lipoprotein-associated phospholipase A₂, GCLM = Glutamate-Cysteine Ligase Modifier Subunit, GR = Glutathione reductase, GPX1 = Glutathione peroxidase, and GCLC = Glutamate-Cysteine Ligase Catalytic Subunit. Bold font indicates significant associations.

**Figure 3 biomedicines-10-01069-f003:**
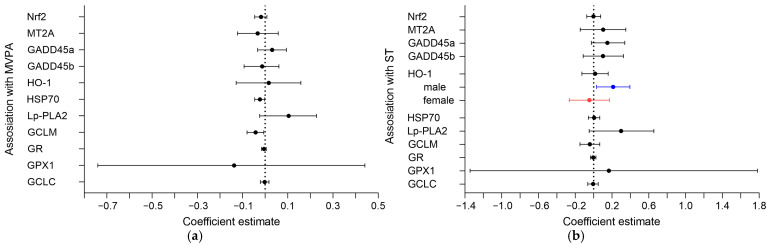
Regression coefficient estimations with confidence intervals showing the association between mean (**a**) MVPA (10-min units) and (**b**) mean ST (10% units) and mRNA expressions (in arbitrary units) of outcome variables. Women with higher MVPA had lower placental HSP70 and GCLM mRNA expression. Women with higher mean ST and pregnant with a male fetus (blue line) had higher placental HO-1 mRNA expression.

## 4. Discussion

The key findings of the present study are: (1) placental GCLM and HSP70 expressions were negatively associated with mean MVPA, averaged over three time points in pregnancy; (2) a sex-specific positive association of mean % ST and changes in % ST with HO-1 gene expression was found in placentas of male, but not female neonates; (3) other placental anti- and pro-oxidative stress markers were not associated with MVPA or ST.

Glutathione (GSH) is a key free radical scavenger with altered levels in different diseases [39]. Glutamate-cysteine ligase (GCL) is the rate-limiting enzyme of GSH synthesis. The enzyme consists of two subunits, a heavy catalytic subunit (GCLC) and a light regulatory subunit (GCLM), both modulating total GCL activity. GCL activity is feedback-inhibited by GSH, thus, higher levels of GSH inhibit GCL. Placental GSH levels are higher in obese than in lean women [40], paralleled by higher superoxide dismutase and catalase activity. This suggests a protective mechanism of the anti-oxidative enzymes when pregnant women are obese [40,41]. Placental GCLC is predominantly present in trophoblasts, the primary tissue exposed to changes in the maternal environment [42]. By inference, we expect the GCLM subunit to be co-located with GCLC in the trophoblasts, and therefore in the first line of defense. We speculate that lower levels of placental GCLM expression, induced by higher MVPA, result from higher anti-oxidative GSH levels. Absence of significant associations of MVPA with Nrf2, a transcription factor upstream of GCLM [32], makes a contributing role of Nrf2 unlikely.

Heat shock proteins (HSPs) are a group of highly conserved proteins that protect the cells from stress-induced damage [43]. HSP transcription responds to a variety of stimuli including pro-oxidants, free radicals, and others. HSP70 attenuates the induction of pro-inflammatory genes, including inducible nitric oxide synthase (NOS), tumor necrosis factor β, interleukin (IL)-1, and IL-6 [44,45,46]. HSPs’ powerful anti-oxidant activity makes them important players in establishing and maintaining pregnancy [47]. In healthy pregnant women, serum levels of HSP70 are lower than in non-pregnant women [48], maintaining maternal immune tolerance. Elevated HSP70 levels have been found in women with preeclampsia, GDM and preterm delivery [49,50,51,52,53,54], which correlated with HbA1c levels and chronic hyperglycemia [51]. In the placenta, the principal location of HSPs is in the syncytiotrophoblast and villous cytotrophoblasts, which makes them excellent candidates for placental protection against oxidative and pro-inflammatory stress. In women with GDM or pre-gestational diabetes, elevated placental levels of HSPs were reported [55,56].

Heme oxygenase (HO), a member of the HSP family designated HSP32, catalyzes NADPH-dependent oxidation of heme to biliverdin, ferrous iron, and carbon monoxide [57]. The HO-1 isoform is induced by oxidative stress and hypoxia and exerts its anti-oxidative and anti-inflammatory effects through the degradation of heme as well as through its downstream byproducts biliverdin and bilirubin [58]. HO-1 deficiency has been linked to a number of pregnancy complications [59], and like GCLC, placental HO-1 is highly expressed in trophoblasts. The positive association of placental HO-1 with mean ST and ST changes in males might indicate that prolonged periods of sedentary behavior contribute to increased oxidative stress in the placenta.

In addition to the positive association of ST with HO-1, we found that HSP70, another member of the HSP family, was negatively associated with average MVPA. The lower HSP70 mRNA levels seem to be a long-term response only found because of averaging MVPA at three time points in pregnancy. Outside of pregnancy, acute MVPA triggers an acute low inflammatory and oxidative state, but when MVPA is repeated regularly, this results in chronic adaptations, which in the long-term lead to lower inflammatory and oxidative stress levels [20]. The associations found in our study do not represent the acute effects of MVPA and ST on placental oxidative stress markers, but reflect the long-term influence of PA and ST levels from early to late pregnancy on these markers. We find associations of PA in the opposite direction from ST: higher MVPA was associated with lower GCML and HSP70, which could be signs of lower inflammation and oxidative stress burden in the placenta. On the other hand, higher ST levels were associated with higher HO-1 levels, demonstrating a higher oxidative state and resulting in higher expression of cellular stress activated factors. This is in line with findings outside of pregnancy (19), and a previous study that detected lower mitochondrial markers of oxidative stress in exercising pregnant women compared to an inactive control group [25]. The comparison with previous studies using animal models is difficult, since the placental response to exercise may vary with species, since exercise training of rats upregulated placental HSPs compared to inactive controls [60].

The effect of fetal sex has been previously detected in placentas, demonstrating a higher adaptivity in placentas of female fetuses [61]. Likewise, the sex interaction in our study represented a significant association of GCLM and MVPA in placentas of female neonates, which was not found in male neonates. This sex-dichotomy in placental function has been well recognized. In contrast, the association of ST with HO-1 was only found in male fetuses. This is in line with finding that associations of ST with maternal glucose and insulin metabolism were mostly driven by women pregnant with a male fetus [62].

### Strengths and Weaknesses

To the best of our knowledge, this is the largest study on the effect of physical activity and sedentary time in pregnancy on placental defense systems. A key strength is the objective, longitudinal measurement of both PA and ST. Most of the previous reports on the associations of PA and ST in pregnancy and oxidative stress markers were based on self-reports and questionnaires, which are subject to bias and errors [28,63]. Furthermore, PA and ST were measured repeatedly at three time points in pregnancy, as opposed to many studies, which have measured PA and ST at only one time point. These longitudinal measurements are important, as we and others [64] report a tendency of women to change their ST and PA levels during the course of pregnancy. Thus, a measurement at one time point is unlikely to represent PA intensity and volume throughout the whole duration of pregnancy. Furthermore, the repeated measurements in our study allowed us to estimate individual changes in PA and ST over the course of pregnancy and include them into our models. A further strength is the inclusion of obese women, mostly (81.3%) Caucasians from eight different study sites in Europe, which enhances representativeness. Therefore, our results can be generalized to Caucasian European women with obesity.

A weakness of this study is a lack of inclusion of lean pregnant women. This precludes testing of a potential different placental response in lean as compared to obese women. In addition, from 436 DALI lifestyle trial participants only 96 had both mRNA quantifications and physical activity measurements of at least two time points to be included in our analysis, however, these women seem to be representative of the total study population. Another limitation is the quantification of only mRNAs, and not protein, which was not possible because of the RCT design and the large sample size, but it limits interpretation of the findings. This calls for inclusion of placental protein measurements in future studies.

## 5. Conclusions

We found a negative relationship between more MVPA and HSP70 mRNA expression in placentas of both sexes and with GCLM expression only in placentas of female fetuses and a positive association between more ST with HO-1 mRNA expression in placentas of male neonates. Together, we speculate that increasing PA and reducing ST might reduce the oxidative stress state in placentas of obese pregnant women. However, studies on effects of PA and ST on placental oxidative stress in women with pathologic pregnancies are rare and more studies are needed. Those studies should quantify the pro- and anti-inflammatory mRNA and protein expressions in the placenta, as well as anti-oxidative capacity of maternal and fetal blood, to better understand their functions in the placenta in relation to PA and ST. ST could also be a noteworthy predictor of health independent of PA and should be included as a separate exposure in future studies.

## Figures and Tables

**Figure 1 biomedicines-10-01069-f001:**
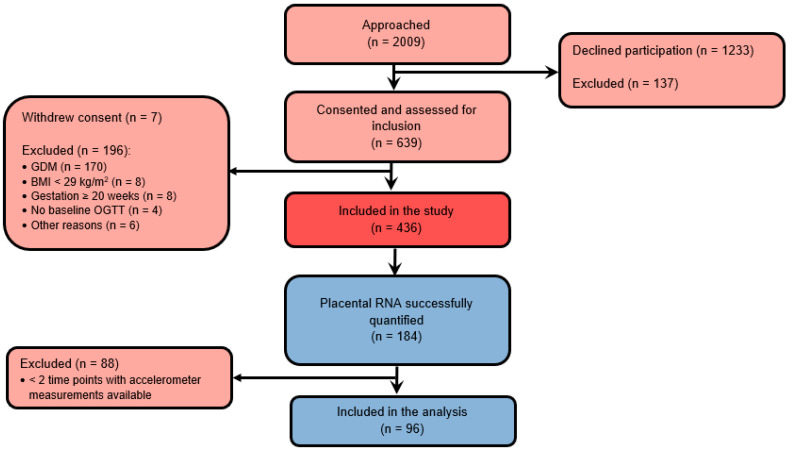
Flow diagram of study participants.

**Figure 2 biomedicines-10-01069-f002:**
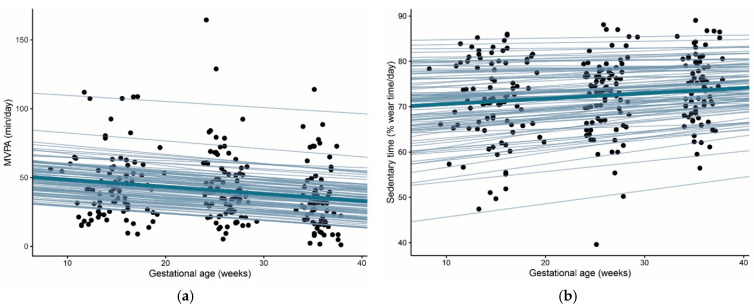
Individual estimates of (**a**) changes in moderate-to-vigorous PA (MVPA) per gestational week and (**b**) changes in % sedentary time (ST) per gestational week. In all women, for MVPA a negative slope, meaning a decrease in MVPA per gestational week was observed. In the majority of women, a positive slope for % ST, meaning an increase in ST, was found.

**Table 1 biomedicines-10-01069-t001:** Characteristics of study sample comparing included and excluded participants.

Maternal Characteristics	n	Included	n	Excluded
Age, years, mean ± SD	96	33.3 ± 5.3	339	31.6 ± 5.3 *
Pre-pregnancy BMI, kg/m^2^, median (IQR)	96	33.7 ± 4.0	339	33.7 ± 4.0
GWG at 35–37 weeks, kg, mean ± SD	92	8.3 ± 5.2	266	7.6 ± 4.4
Nulliparity, count (%)	96	50 (52.1%)	339	164 (48.4%)
High education, count (%)	96	57 (59.4%)	339	181 (53.4%)
European descent, count (%)	96	78 (81.3%)	339	299 (88.2%)
Smoking, count (%)	96	11 (11.5%)	317	44 (13.9%)
GDM total, count (%)	93	32 (34.4%)	268	101 (37.7%)
**Neonatal characteristics**				
Birthweight, g, mean ± SD	96	3604.6 ± 499.8	297	3448.6 ± 554.8 *
Placenta weight, g, mean ± SD	91	634.5 ± 147.8	207	651.0 ± 376.3
Gestational age at birth, weeks, mean ± SD	96	40.0 ± 1.2	297	39.4 ± 2.4 *
Female sex, count (%)	96	44 (45.8%)	300	155 (51.7%)
Caesarean section, count (%)	95	26 (27.4%)	288	101 (35.1%)

* Represents a significant difference between included and excluded participants (*p* < 0.05). GWG = gestational weight gain, GDM = gestational diabetes mellitus, SD = standard deviation, and IQR = interquartile range.

**Table 2 biomedicines-10-01069-t002:** Moderate-to-vigorous physical activity (MVPA) and sedentary time (ST) at three time points in pregnancy.

	<20 Weeks n = 87	24–28 Weeks n = 83	35–37 Weeks n = 75
MVPA, min/day, median (IQR)	41.9 (26.0–56.4)	38.9 (25.9–55.4) *	31.4 (18.2–43.0) **
Sedentary time, % of wear time, mean ± SD	71.4 ± 9.1	72.5 ± 8.2 *	74.1 ± 7.2 *

* *p* < 0.05 compared to <20 weeks. ** *p* < 0.05 compared to 24–28 weeks and <20 weeks. MVPA = moderate-to-vigorous physical activity, IQR = interquartile range, and SD = standard deviation.

## Data Availability

The raw data supporting the conclusions of this manuscript will be made available by the authors, without undue reservation, on request to the corresponding author.

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
