# Peer review of "Physical Activity and Sedentary Time in Pregnancy: An Exploratory Study on Oxidative Stress Markers in the Placenta of Women with Obesity"

_biomedicines, 2022, doi:10.3390/biomedicines10051069_

Round 1

Reviewer 1 Report

The rationale for studying the specific mRNA transcript is not clear. However, the primary concern of the experiment design is the lack of lean BMI subjects, even though the study was carried out across the gestation time points using the obese BMI subjects.    

Author Response

  • The rationale for studying the specific mRNA transcript is not clear.

We thank the reviewer for the comments. As not much is known on whether and how physical activity influences placental oxidative stress state, this was designed as an explorative study. We aimed to include multiple targets that have proven useful markers to identify oxidative stress, or are involved in oxidative stress response in the placenta or in other tissues (mentioned on page 2, lines 46-48). The specific targets were chosen either based on our own previous results or from literature. Nrf2 is a transcription factor acting as master regulator of its downstream effectors HSP, HO, GADD45 and Lp_PLA2. Components of the glutathione system are ubiquitous key actors in the intracellular oxidative stress systems. We have added this information in the Methods section (page 4, lines 4-6).

  • However, the primary concern of the experiment design is the lack of lean BMI subjects, even though the study was carried out across the gestation time points using the obese BMI subjects.    

Obese pregnant women have a high risk of developing GDM, which is a condition associated with oxidative stress. Should this oxidative stress be superimposed by physical activity, which may also entail oxidative stress, then the limit of placental adaptive responses may be exceeded with adverse consequences for the fetus. Hence, we have used this group of obese women with the highest probability to identify changes in the targets. However, we agree that inclusion of a lean group of pregnant women had strengthen the study. We agree this as a limitation and have addressed this in the discussion (page 9, line 52).

Reviewer 2 Report

The work by Zafaranieh et al., provides interesting results regarding the effects of physical activity and sedentary time during gestation on oxidative stress markers in the placenta. The study is well designed and conducted, and the manuscript is very well written.

There are only minor points that could be addressed:

  • It is explained that women were randomized into four groups receiving different counselling: for Healthy Eating, Physical Activity, Healthy Eating + Physical Activity, and usual care. However, it is stated that the analyses were made in all participants combined into one cohort, without considering the type of intervention. Have the authors examined potential differences regarding the type of intervention? Beside of the interest of the observational analysis, the effect of intervention could be of interest.
  • Have the authors considered the potential effects of exercise in the offspring, e.g. regarding body weight at birth? This is of interest considering the potential interest of maternal exercise during gestation to prevent adverse birth outcomes and possible childhood obesity.

Author Response

  • It is explained that women were randomized into four groups receiving different counselling: for Healthy Eating, Physical Activity, Healthy Eating + Physical Activity, and usual care. However, it is stated that the analyses were made in all participants combined into one cohort, without considering the type of intervention. Have the authors examined potential differences regarding the type of intervention? Beside of the interest of the observational analysis, the effect of intervention could be of interest.

We thank the reviewer for the useful comments. We were not able to investigate effects of the intervention as information on the mRNA markers was only available for a subset of participants, and therefore the randomization groups were not intact anymore. We did look, however, at differences between intervention groups and did not find any effects of the interventions on mRNA markers (data not shown in the paper, given the small sample size). 

All analyses were adjusted for potential intervention effects, however only for healthy eating (see page 4, line 53). As we were looking at physical activity as main exposure, we did not adjust for the physical activity intervention.

  • Have the authors considered the potential effects of exercise in the offspring, e.g. regarding body weight at birth? This is of interest considering the potential interest of maternal exercise during gestation to prevent adverse birth outcomes and possible childhood obesity.

The intervention did not have an effect on birthweight (paper Simmons 2017, PMID: 27935767), but there was a significant effect of the combined healthy eating and physical activity intervention on neonatal adiposity (paper van Poppel 2019, PMID: 30840112) and cord blood leptin. We have mentioned this in the paper (page 2, line 23). We also recently investigated the longitudinal association of physical activity with neonatal adiposity and again found that physical activity was associated with reduced neonatal adiposity, but not birthweight (paper under review).